# Human Deoxycytidine Kinase Is a Valuable Biocatalyst for the Synthesis of Nucleotide Analogues

Katja F. Hellendahl [1], Sarah Kamel [1,2], Albane Wetterwald [2,3], Peter Neubauer [1]  and Anke Wagner [1,2,*]

[1] Faculty III Process Sciences, Institute of Biotechnology, Technische Universität Berlin, Ackerstr. 76, 13355 Berlin, Germany; k.hellendahl@campus.tu-berlin.de (K.F.H.); sarah.kamel@aucegypt.edu (S.K.); peter.neubauer@tu-berlin.de (P.N.)
[2] BioNukleo GmbH, Ackerstr. 76, 13355 Berlin, Germany; albane.wetterwald@gmail.com
[3] Ecole Nationale Supérieure d'Agronomie et des Industries Alimentaires, 2 Avenue de la Forêt de Haye, 54505 Vandœuvre-lès-Nancy, France
* Correspondence: anke.wagner@tu-berlin.de; Tel.: +49-30-314-72183

**Abstract:** Natural ribonucleoside-5′-monophosphates are building blocks for nucleic acids which are used for a number of purposes, including food additives. Their analogues, additionally, are used in pharmaceutical applications. Fludarabine-5´-monophosphate, for example, is effective in treating hematological malignancies. To date, ribonucleoside-5′-monophosphates are mainly produced by chemical synthesis, but the inherent drawbacks of this approach have led to the development of enzymatic synthesis routes. In this study, we evaluated the potential of human deoxycytidine kinase (*Hs*dCK) as suitable biocatalyst for the synthesis of natural and modified ribonucleoside-5′-monophosphates from their corresponding nucleosides. Human dCK was heterologously expressed in *E. coli* and immobilized onto Nickel-nitrilotriacetic acid (Ni-NTA) superflow. A screening of the substrate spectrum of soluble and immobilized biocatalyst revealed that *Hs*dCK accepts a wide range of natural and modified nucleosides, except for thymidine and uridine derivatives. Upon optimization of the reaction conditions, *Hs*dCK was used for the synthesis of fludarabine-5´-monophosphate using increasing substrate concentrations. While the soluble biocatalyst revealed highest product formation with the lowest substrate concentration of 0.3 mM, the product yield increased with increasing substrate concentrations in the presence of the immobilized *Hs*dCK. Hence, the application of immobilized *Hs*dCK is advantageous upon using high substrate concentration which is relevant in industrial applications.

**Keywords:** human deoxycytidine kinase; Ni-NTA sepharose; activity screening; nucleoside analogue; cladribine; clofarabine; fludarabine

---

## 1. Introduction

Modified nucleosides and nucleotides are important small molecules in molecular, biological and pharmaceutical applications. Since they are involved in the same metabolic pathways as endogenous nucleosides and nucleotides, they act as antimetabolites, thus, exerting useful pharmacological action [1]. Most of the known nucleoside analogue drugs are activated in vivo to their respective active forms (nucleoside-5′-diphosphate or nucleoside-5′-triphosphate), nevertheless, in vivo activation is often insufficient [2]. Therefore, in a number of approaches, bio-reversible protected ribonucleoside-5′-monophosphates (NMPs), such as the phosphoramidate prodrug sofosbuvir were administered to overcome the first activation step in vivo [3–6].

To date, nucleotides are mainly produced by chemical methods like the Yoshikawa protocol or the Ludwig-Eckstein method [7]. A common disadvantage of these multistep chemical synthesis reactions is a limited regio- and stereo-selectivity leading to a need for protection/deprotection steps. Several side products are produced, which in return complicates the purification process. The overall process is laborious, and often only moderate product yields are achieved. Furthermore, nucleotide products harboring sensitive functional groups, such as a triazide ring cannot be synthesized at all in a chemical approach, as they cannot withstand the harsh conditions [8].

Despite the progress made in the optimization of chemical synthesis routes, enzymatic or chemo-enzymatic methods are often a good alternative [9]. Compared to chemical synthesis routes, biotransformation usually does not require protection or deprotection steps and lead to high final product yields, due to the high regio- and stereo-selectivity of the enzymes. Deoxyribonucleoside kinases were described as suitable biocatalysts for the synthesis of nucleotide analogues [10,11]. As an example, the deoxynucleoside kinase (dNK) from the fruit-fly *Drosophila melanogaster* was shown to have a wide substrate spectrum and to use 2′-deoxy-ribonucleosides, arabinosyl-uracil, arabinosyl-adenine and fludarabine as substrates [11].

Human deoxycytidine kinase (*Hs*dCK) is another interesting biocatalyst for the synthesis of nucleotide analogues. In vivo, it plays an important role in maintaining the homeostasis of the 2′-deoxy-ribonucleotide pool within the cell. Using adenosine triphosphate or uridine triphosphate as a phosphate donor, it catalyzes the 5′-phosphorylation of deoxycytidine, deoxyadenosine, and deoxyguanosine [12–15]. Furthermore, dCK activity is essential for the activation of a wide variety of important therapeutic prodrugs, such as the antiviral agent lamivudine (3TC), for the treatment of HIV infection [16], or antineoplastic agents like cladribine [17] (Scheme 1).

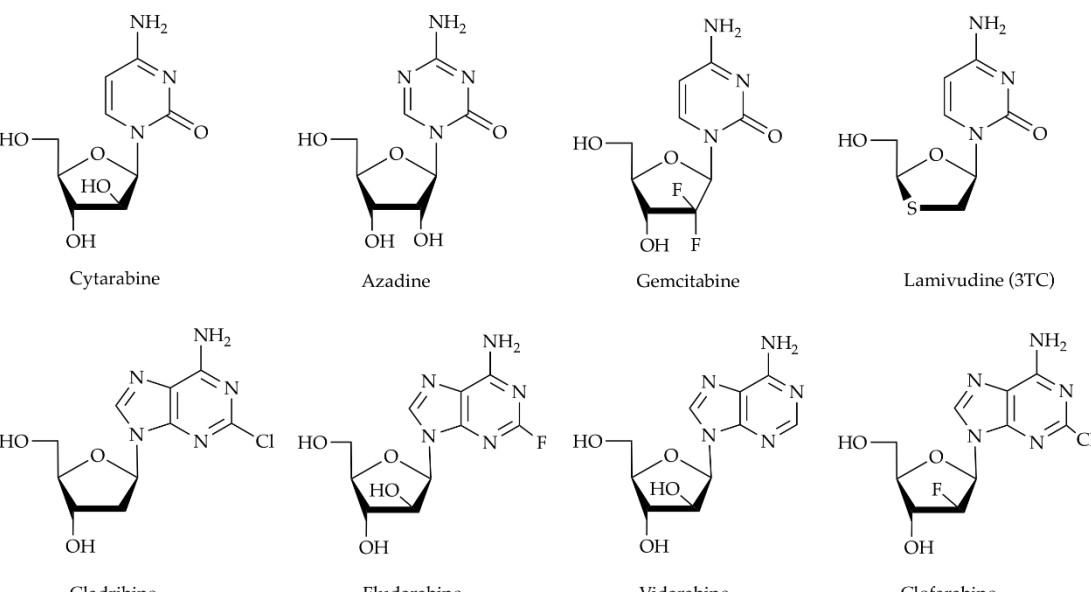

**Scheme 1.** Nucleoside analogues accepted as substrates of human deoxycytidine kinase (*Hs*dCK).

Both whole immobilized cells and purified enzymes have been used for the synthesis of NMP analogues [18,19]. Immobilization techniques help to stabilize biocatalysts under harsh conditions (e.g., high temperature, extreme pH, solvents) [19]. As the immobilized biocatalysts can be used repeatedly, the processes get more cost-effective, which is an important consideration for industrial applications. Using dNK and deoxyadenosine kinase (dAK) of *Drosophila melanogaster* and *Dictyostelium discoideum*, respectively, fludarabine-5′-monophosphate (2F-AraAMP) was synthesized from fludarabine (2F-AraA) with a high conversion percentage (> 90%) [10,11].

In this study, the chemo-enzymatic synthesis of NMP analogues from nucleosides using soluble and immobilized *Hs*dCK is demonstrated (Scheme 2). *Hs*dCK was chosen as it is a well-studied

enzyme showing a wide substrate spectrum. However, it is potential for industrial application was not studied so far. Nickel-nitrilotriacetic acid (Ni-NTA) superflow was applied as the support, since the applied enzymes bear an N-terminal His-tag and it is described as a gentle way of immobilization. Enzyme immobilization did not influence the substrate spectrum of *Hs*dCK. It was shown that *Hs*dCK is a valuable biocatalyst for the synthesis of NMP analogues. Compared to the soluble biocatalyst, immobilized *Hs*dCK showed increased efficiency with increasing substrate concentration.

Cladribine: $R_1$= Cl, $R_2$ = H
Clofarabine: $R_1$= Cl, $R_2$=Arabino-F
Fluadarabine: $R_1$ = F, $R_2$ = Arabino-OH

**Scheme 2.** The enzymatic reaction catalyzed by *Hs*dCK.

## 2. Results

### 2.1. Production of Soluble and Immobilized HsdCK

After expression in a fed-batch medium in shake flasks, *Hs*dCK was either immobilized onto Ni-NTA superflow resin or produced as a soluble enzyme by elution from Ni-NTA superflow material (Figure 1A, Figure S1). Three independent enzyme immobilizations/purification experiments were performed with five replicas for each experiment. While three replicates were used as immobilized biocatalysts, two replicas were used for protein elution to estimate the amount of bound enzyme and to be later used as a soluble biocatalyst for the synthesis of nucleotide analogues.

In preliminary experiments, the amount of cell pellet loaded to the matrix was optimized. When loading the lysate prepared of 0.5 g of cell pellet to 100 µl of the matrix and using 500 µl of elution buffer, the protein concentration in each elution fraction was below 0.5 mg mL$^{-1}$ (Figure S2). This was insufficient for further experiments. Hence, lysate derived from 1.5 g of cell pellet was loaded, and the volume of elution buffer was reduced to 100 µL. By using this protocol protein, the concentrations strongly increased.

Protein concentrations were measured by NanoDrop at 280 nm. Protein concentration in the soluble fraction of the cell extract was around 46 mg mL$^{-1}$. In the elution fractions, protein concentrations between 1 mg mL$^{-1}$ to 5.5 mg mL$^{-1}$ were determined (Figure 1B). Highest protein concentrations were observed in the first elution fraction for all immobilization experiments. The total amount of protein bound to 100 µl of the matrix was between 0.5 (experiment 3) to 1.2 mg (experiment 1). The amount of bound protein was comparable within the replicates of one immobilization (Figure 1C). However, differences were observed between the different immobilization experiments. This observation is probably caused by using different cell pellets for each immobilization experiment. Therefore, it was necessary to determine the amount of bound protein for each immobilization experiment.

Samples obtained from the elution fractions were analyzed by SDS-PAGE and expression of *Hs*dCK in the lysate of cell extract was verified (Figure 1B). For all immobilization experiments, purified *Hs*dCK was detected in the elution fractions, and only a few and comparable impurities were observed. Purity was estimated to be approximately 90%.

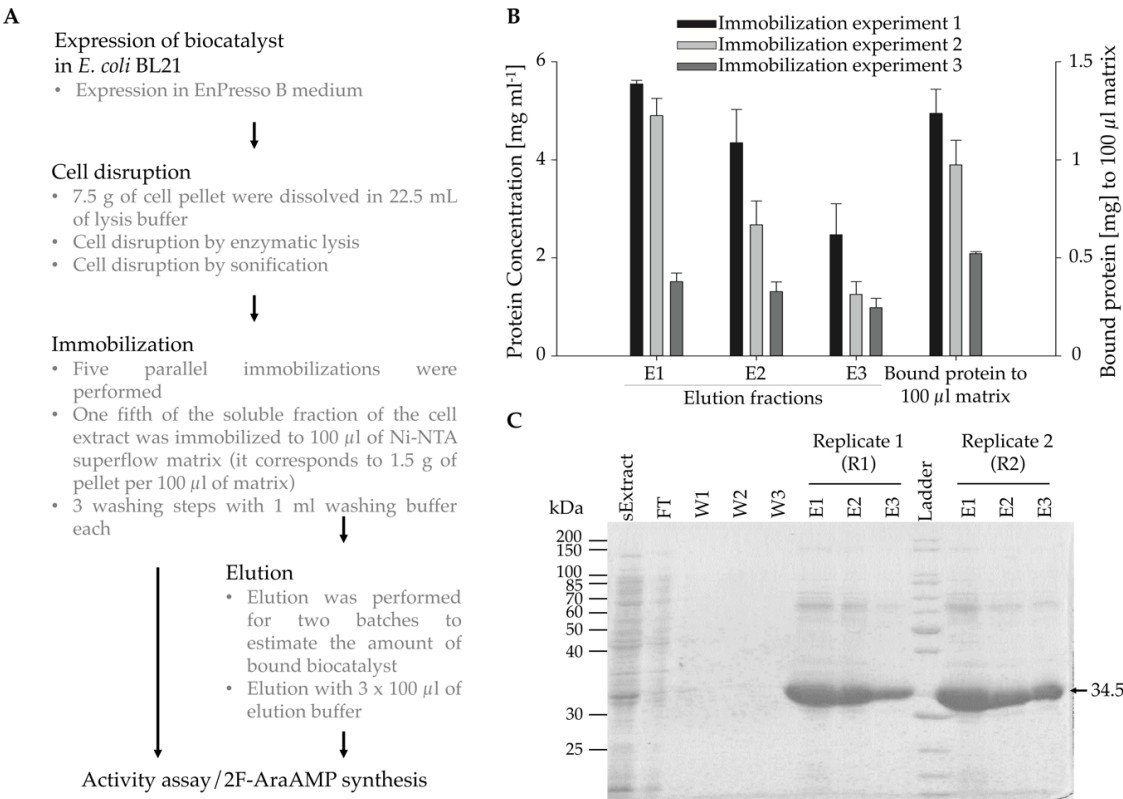

**Figure 1.** Immobilization of *Hs*dCK to Ni-NTA superflow via affinity interaction. (**A**) The strategy used for each immobilization experiment with *Hs*dCK. After expression in Enpresso B medium, cells were disrupted by enzymatic and mechanical lysis. Each immobilization experiment was performed with five replicas. Two replicas were used for protein elution. (**B**) Protein concentrations in the elution fractions were determined by NanoDrop measurements. The amount of the totally bound protein was calculated. Standard deviations were calculated from two replicas of the eluted protein. (**C**) The purification process was validated by SDS-PAGE. The two replicas of immobilization experiment 1 are presented as an example. The target protein is highlighted by an arrow (34.5 KDa). sExtract—soluble fraction of the cell extract; FT—flow through; W1-W3—washing fractions; E1-E3—elution fractions.

## 2.2. Enzymatic Synthesis of NMP Analogues

Soluble and immobilized *Hs*dCK were tested for their substrate spectrum with a variety of natural and modified nucleosides (Figure 2). The screening assay was applied as a qualitative measure. Reactions were analyzed after 20 h to guarantee product formation even with poor substrates. The stability of the soluble and immobilized biocatalyst and the phosphate donor (GTP) was studied in advance. After incubating *Hs*dCK for 24 h at 37 °C, no influence on deoxycytidine-5′-monophosphate formation was observed using reaction times of 24 h (Figure S3A). No impact on the stability of the phosphate donors was observed by incubating them for 24 h at 37 °C (Figure S3B,C).

Using soluble and immobilized *Hs*dCK, natural deoxynucleosides were converted to their equivalent 5′-monophosphate except for thymidine and deoxyuridine (Figure 2A). These two were not accepted as a substrate by *Hs*dCK. While the percentage of conversion was approximately 100% using soluble *Hs*dCK, it was between 20% to 40% with the immobilized biocatalyst.

Additionally, a wide variety of modified nucleosides were accepted as substrates, including cytarabine, cladribine, fludarabine, clofarabine and gemcitabine (Figure 2B). With arabinosyl-adenine, azacytidine and lamivudine, lower product formation was observed. Nucleosides harboring a thymine or modified uracil base (zidovudine, stavudine and iodoxuridine) were not accepted by heterologously expressed *Hs*dCK. In general, the same trend was observed with both the immobilized and the soluble biocatalyst; however, conversion with immobilized enzyme was mainly lower (Figure 2).

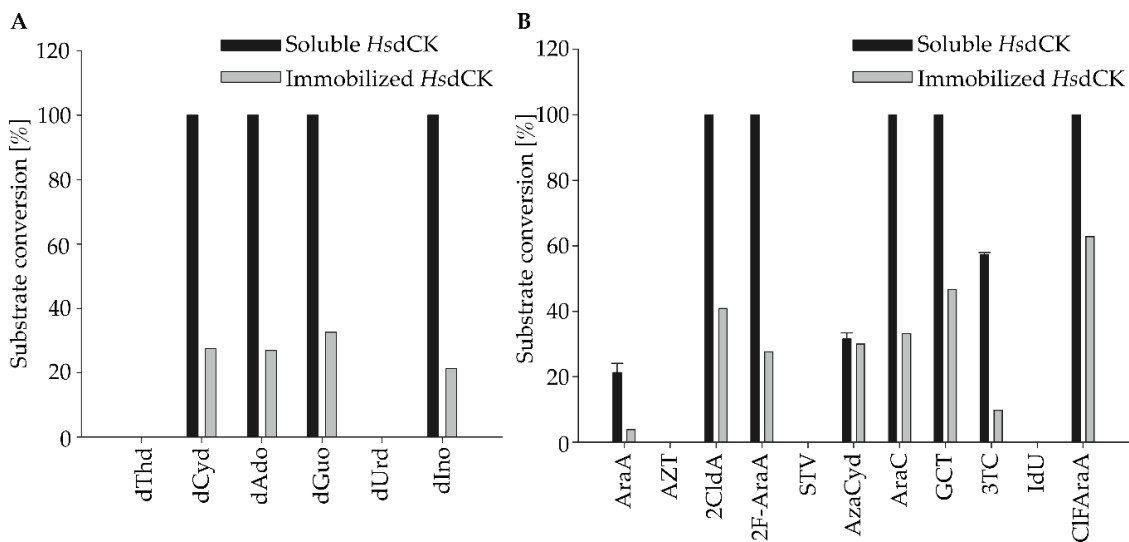

**Figure 2.** Substrate spectrum of human dCK with natural (**A**) and modified (**B**) nucleosides as substrates. Reactions were performed in 2 mM potassium phosphate buffer pH 7 with 10 mM MgCl$_2$, 0.4 mM of GTP and 0.333 mM of substrate in a reaction volume of 0.5 mL. A reaction temperature of 37 °C was used. 0.01 mg mL$^{-1}$ of the enzyme was added to the reaction. The reactions were analyzed after a reaction time of 20 h. dThd—deoxythymidine, dCyd—deoxycytidine, dAdo—deoxyadenosine, dGuo—deoxyguanosine, dUrd—deoxyuridine, dIno—deoxyinosine, AraA—arabinosyladenine (vidarabine), AZT—zidovudine, 2CldA—2-chlorodeoxyadenosine (cladribine), 2F-AraA—2-fluoroarabinoadenosine (fludarabine), STV—stavudine, Azacyd—5-azacytidine (azadine), GCT—gemcitabine, AraC—arabinosylcytosine (cytabrabine), 3TC—lamivudine, IdU—idoxuridine, ClFaraA—2-chloro-2′-fluoroarabinoadenosine (clofarabine). Mean values and standard deviations were calculated from duplicate reactions of pooled replicates of immobilization experiment 3.

The impact of immobilization of *Hs*dCK to Ni-NTA sepharose was studied in more detail. Therefore, specific activities and product formation were determined for cladribine, fludarabine and clofarabine phosphorylation reactions. When analyzing specific activities determined for cladribine, fludarabine and clofarabine, higher values were observed for the soluble enzyme compared to the immobilized enzyme (Table 1). Comparable values were observed for cladribine and fludarabine. Specific activity was lowest with clofarabine as substrate. Matrix-bound-enzyme activity was between 0.76 U mL$^{-1}$ (clofarabine) to 1.15 U mL$^{-1}$ (fludarabine) (Table S1).

**Table 1.** Phosphorylation of nucleoside analogues cladribine, fludarabine and clofarabine by *Hs*dCK. Specific activity and percentage of conversion were determined. Reactions were performed in 2 mM potassium phosphate buffer (pH 7) with 10 mM MgCl$_2$, 0.4 mM of GTP and 0.333 mM of substrate in a reaction volume of 1 mL. A reaction temperature of 37 °C was used. 0.01 mg mL$^{-1}$ of the enzyme was added to the reaction. The reactions were monitored over a period of 60 min. Mean values and standard deviations were calculated for each of the three replicas of immobilization experiment 3.

| Substrate | Enzyme | Specific Activity [U mg$^{-1}$] | Retained Activity [%] | Conversion [%] |
|---|---|---|---|---|
| Cladribine | immobilized | 0.22 (±0.06) | 67 | 52 (±1) |
| | soluble | 0.33 (±0.04) | | 62 (±3) |
| Fludarabine | immobilized | 0.24 (±0.18) | 41 | 25 (±6) |
| | soluble | 0.58 (±0.10) | | 46 (±7) |
| Clofarabine | immobilized | 0.16 (±0.03) | 84 | 36 (±6) |
| | soluble | 0.19 (±0.03) | | 38 (±2) |

The impact of immobilization on the percentage of conversion was more pronounced for cladribine and fludarabine, while differences were minor for clofarabine (Table 1). For the soluble *Hs*dCK, percentage of conversion increased in the order: clofarabine (62%), fludarabine (46%), cladribine (38) (Table 1). With the immobilized biocatalyst, the highest percentage of conversion was obtained with cladribine as well. However, in contrast to the soluble enzyme, conversion was higher for clofarabine compared to fludarabine.

The retained activity was determined based on the specific activity determined for the soluble and the immobilized *Hs*dCK (Table 1). Retained activity differed for the different substrates tested. Retained activity was lowest for fludarabine (41%) and highest for clofarabine (84%).

## 2.3. Optimization of the Reaction Conditions

As fludarabine-5´-phosphate is an approved drug used to treat hematological malignancies; we intended to optimize the reaction conditions for its synthesis. Using the standard reaction conditions and a reaction duration of 1 h, product yields of 21% and 54% were observed using immobilized and soluble *Hs*dCK as biocatalyst, respectively (Table 2). In order to increase the product yield, the influence of the phosphate donor, the $MgCl_2$ concentration and the ratio of phosphate donor to the substrate was analyzed.

**Table 2.** Phosphorylation of fludarabine using different reaction conditions. The impact of phosphate donor and varying phosphate donor and $MgCl_2$ concentration on the product formation was studied. Reactions were performed in 2 mM potassium phosphate buffer (pH 7) in a reaction volume of 1 mL. A reaction temperature of 37 °C was used. 0.01 mg mL$^{-1}$ of the enzyme was added to the reaction. The reactions were monitored over a period of 60 min. Mean values and standard deviations were calculated from two independent experiments performed with pooled replicas of immobilization experiment 3.

| Fludarabine (mM) | Phosphate Donor | $MgCl_2$ (mM) | Enzyme | Conversion % (Standard Deviation) |
|---|---|---|---|---|
| Standard reaction condition | | | | |
| 0.333 | GTP—0.4 mM | 10 mM | immobilized | 21 (0.6) |
| 0.333 | GTP—0.4 mM | 10 mM | soluble | 54 (0.4) |
| Impact of phosphate donor | | | | |
| 0.333 | ATP—0.4 mM | 10 mM | immobilized | 35 (0.8) |
| 0.333 | ATP—0.4 mM | 10 mM | soluble | 53 (0.7) |
| 0.333 | UTP—0.4 mM | 10 mM | immobilized | 3 (0.8) |
| 0.333 | UTP—0.4 mM | 10 mM | soluble | 13 (0.9) |
| Impact of ratio of substrate to phosphate donor | | | | |
| 0.4 | GTP—0.4 mM | 10 mM | immobilized | 21 (0.1) |
| 0.4 | GTP—0.4 mM | 10 mM | soluble | 49 (1.6) |
| 0.2 | GTP—0.4 mM | 10 mM | immobilized | 16 (1.4) |
| 0.2 | GTP—0.4 mM | 10 mM | soluble | 58 (2.1) |
| 0.1 | GTP—0.4 mM | 10 mM | immobilized | 12 (0.9) |
| 0.1 | GTP—0.4 mM | 10 mM | soluble | 75 (0.3) |
| Impact of $MgCl_2$ concentration | | | | |
| 0.333 | GTP—0.4 mM | 0.4 mM | immobilized | 18 (0.5) |
| 0.333 | GTP—0.4 mM | 0.4 mM | soluble | 51 (1) |
| 0.333 | GTP—0.4 mM | 1 mM | immobilized | 19 (0.7) |
| 0.333 | GTP—0.4 mM | 1 mM | soluble | 51 (0.1) |
| 0.333 | GTP—0.4 mM | 2 mM | immobilized | 22 (1.1) |
| 0.333 | GTP—0.4 mM | 2 mM | soluble | 53 (0.7) |

Ericksson and Wang showed that uridine triphosphate (UTP) is likely to be the major intracellular phosphate donor for *Hs*dCK [20]. Hence, it was expected that UTP is a better phosphate donor compared

to adenosine triphosphate (ATP). We studied the impact of guanosine triphosphate (GTP), ATP and UTP on the conversion of fludarabine using heterologously expressed *Hs*dCK. While the conversion percentages were comparable for both ATP and GTP, fludarabine-5'-monophosphate formation was significantly less using UTP as the phosphate donor. Only 13% and 3% conversion percentages were observed using soluble and immobilized *Hs*dCK, respectively. Hence, conversion percentages were reduced by a factor of 4% to 7%. As comparable results were obtained for both GTP and ATP, further experiments were performed with GTP.

The molar ratio of fludarabine and GTP was analyzed further. If the soluble protein is considered first, a higher molar ratio of GTP led to increased conversion of fludarabine (Table 2). The lowest conversion of 49% was achieved with an equimolar concentration of fludarabine and GTP, when using a 4-fold excess of GTP, the conversion was 75% after a reaction time of one hour. In contrast, with the immobilized biocatalyst, we found that the percentage of conversion decreased with increasing excess of GTP. While the conversion was 21% with molar ratios of fludarabine to GTP of 1:1 and 1:1.2, the conversion was only 12% with a 4-fold excess of GTP (Table 2). For up-scaling experiments, a molar ratio of 1:1.2 was used to meet the requirements for reactions with both soluble and immobilized *Hs*dCK.

Magnesium is essential for catalysis involving nucleotides [21]. Therefore, the impact of different $MgCl_2$ concentrations on the conversion of fludarabine was studied. No significant impact was observed using $MgCl_2$ concentrations between 0.4 to 10 mM (Table 2). Thus, for further experiments, an $MgCl_2$ concentration of 10 mM was applied.

When comparing the described data with the initial enzyme screening, reaction time has the strongest impact on the final product yield. In the screening assay, 100% of fludarabine conversion was determined using a reaction time of 20 h.

## 2.4. Reusability of Immobilized HsdCK

A huge advantage of biocatalyst immobilization is the possibility of re-using, thus, increase the efficiency of the catalytic process. The durability of the immobilized *Hs*dCK was tested (Figure 3). Three consecutive reactions were performed with the same batch of immobilized enzyme. Our results have shown a slight decrease (17%) of enzyme activity between reaction 1 and 2 as could be denoted by the decrease in the formation of fludarabine-5´-monophosphate (Figure 3, Figure S4). The relative formation of fludarabine-5'-monophosphate formation was decreased by approximately 80% in the third repetition as compared to the first reaction (Figure 3). Losses in enzyme activity after each re-use are probably caused by detachment of the protein from the matrix which is increasing over time.

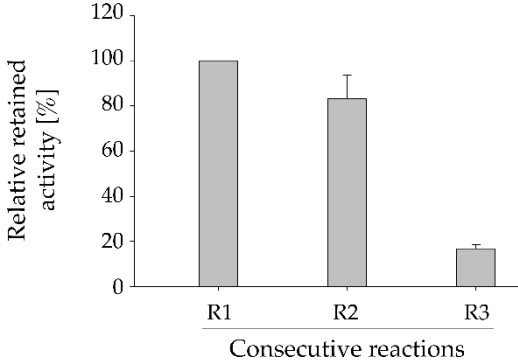

**Figure 3.** Reusability of the immobilized *Hs*dCK for the synthesis of fludarabine-5´-monophosphate. Relative retained activity in three consecutive reactions was determined. Reactions were performed in 2 mM potassium phosphate buffer (pH 7) with 10 mM $MgCl_2$, 0.4 mM of GTP and 0.333 mM of substrate in a reaction volume of 1 mL using a reaction time of 3 h for each reaction. Mean values and standard deviations were calculated from triplicate reactions performed with pooled replicas of immobilization experiment 3.

### 2.5. Impact of Increasing Substrate Concentrations on HsdCK-Catalyzed Reactions

For industrial application, it is very important to test the possibility to upscale the enzymatic reaction. Therefore, increasing substrate concentrations (3- to 36-fold) were applied in combination with increasing biocatalyst concentrations in *Hs*dCK catalyzed reactions. Compared to the standard reaction conditions, the percentage of formation of fludarabine-5´-monophosphate was decreased upon using soluble *Hs*dCK and increased with immobilized *Hs*dCK (Table 2, Figure 4). In reactions with soluble enzyme, after 18 h, fludarabine-5´-monophosphate was formed with 71%, 69%, and 60% upon using 1 mM (3 folds), 6.5 mM (19.5 folds) and 12 mM (36 folds) fludarabine, respectively, as opposed to 100% formation using the standard conditions. Whereas, with the immobilized *Hs*dCK catalyzed reactions, the formation of fludarabine-5´-monophosphate increased from 27% with the standard reaction conditions to 47%, 37% and 55%, respectively. Using a substrate concentration of 12 mM product concentrations of 8.7 ($\pm$0.2) mM and 6.0 ($\pm$0.8) mM were measured (Table S2) for the soluble and immobilized *Hs*dCK, respectively. The application of the immobilized biocatalyst is advantageous when higher substrate concentrations are applied.

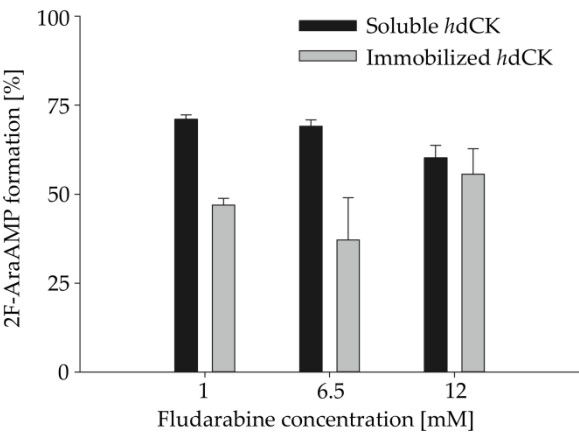

**Figure 4.** Impact of increasing substrate and enzyme concentrations on the synthesis of fludarabine-5´-monophosphate. Fludarabine concentration was increased to 3 fold (1 mM), 19.5 fold (6.5 mM) and 36 fold (12 mM) compared to the standard reaction conditions using both free and immobilized *Hs*dCK. Product formation was detected by HPLC analysis. Reactions were performed in 2 mM potassium phosphate buffer (pH 7) and 10 mM $MgCl_2$ in a volume of 100 $\mu$l and using a reaction time of 18 h. Enzyme concentrations were increased to 0.03 mg mL$^{-1}$, 0.19 mg mL$^{-1}$ and 0.36 mg mL$^{-1}$ with the increase of the substrate concentrations respectively. Mean values and standard deviations were calculated from two independent experiments performed with pooled replicas of immobilization experiment 3.

## 3. Discussion

Human dCK is a suitable biocatalyst for the synthesis of natural and modified nucleoside-5'-monophosphates. After immobilization to Ni-NTA superflow, the substrate spectrum of the immobilized *Hs*dCK was not affected; however, the conversion percentages and the enzyme-specific activity was decreased in comparison to the soluble enzyme. Although, ionic adsorption to charged supports is described to be a mild way of enzyme immobilization [19], our results showed a decreased substrate conversion percentages using low substrate concentrations. This might be explained by mass-transfer limitations of the substrate to contact the enzyme [22] especially as the applied Ni-NTA sepharose is a gel-like matrix. Mass-transfer limitations are widely studied applying immobilized enzymes and served as an explanation for the lowered enzyme-specific activity [23–26]. Different mechanisms can be responsible for these limitations. Either active sites are not accessible anymore due to immobilization or there is a diffusional limitation which leads to gradient formation [27]. In the latter case, lower product concentrations within the matrix are observed. Interestingly, in this

study the impact of immobilization on the percentage of conversion differed between the substrates applied (Table 1). This might be explained by differences in the chemistry of the compounds, which is influencing the diffusion of the substrates into the matrix.

While mass-transfer limitations negatively influence final product yields at low substrate concentrations (Table 1), this is beneficial when higher and/or inhibitory substrate concentrations are applied (Figure 4). For *Hs*dCK, it was shown that dCyd is causing enzyme inhibition [28] which might explain the decreased product formation with increasing substrate concentration (Figure 4). A beneficial effect of immobilized biocatalysts at increasing substrate concentrations was also observed using *Dm*dNK [11] or invertase of *Candida utilis* [29] as biocatalyst. Slow diffusion of the substrate into the matrix also served as an explanation for the reduced inhibition observed with the immobilized biocatalyst [29].

Although reusability of the immobilized enzyme is a very attractive advantage of immobilization, yet, a challenge is the stable attachment of the biocatalyst to the matrix. In this study, loss of activity was observed in consecutive reactions. This is in good accordance with previous observations where protein leach from the matrix was observed when non-covalent immobilization was applied [30]. Due to the limitations of non-covalent binding, post-immobilization by cross-linking and covalent binding were developed to avoid desorption of enzyme under more harsh conditions [19,31–33]. As an example, Serra and colleagues compared different strategies to cross-link with oxidized dextran [33]. The treatment with dextran aldehyde efficiently prevented desorption of the enzyme from the carrier. Hence, the combination of ionic interaction with cross-linking might be a suitable option to ensures that *Hs*dCK is tightly bound to the matrix and that the product is not contaminated with enzymes.

For the native *Hs*dCK, UTP was described to be the preferred phosphate donor in comparison to ATP [20,28]. Using the heterologously expressed variant of *Hs*dCK higher product yields were obtained with GTP and ATP compared to UTP (Table 2). Therefore, GTP was applied as a phosphate donor in this study. Kinetic studies with the heterologously expressed protein are, however, needed to study inhibitory effects by increasing substrate concentrations in the presence of UTP, ATP and GTP in more detail [28].

## 4. Materials and Methods

### 4.1. General Information

All chemicals and solvents were of analytical grade or higher and purchased, if not stated otherwise, from Sigma-Aldrich (Steinheim, Germany), Carl Roth (Karlsruhe, Germany), TCI Deutschland (Eschborn, Germany), Carbosynth (Berkshire, UK) or VWR (Darmstadt, Germany). All nucleobases, nucleosides and nucleotides were purchased from Carbosynth (Berkshire, UK). 2F-AraA was purchased from Alfa-Aesar (Kandel, Germany) with a purity of 98% and 2F-Ara-AMP from Sigma-Aldrich (Steinheim, Germany). Ni-NTA superflow was obtained from Qiagen (Hilden, Germany). Ni-NTA was obtained as a 50% solution. The matrix has a gel-like appearance. HPLC analyses were carried out with an Agilent 1200 series system equipped with an Agilent DAD detector using a Phenomenex (Torrance, CA, USA) reversed-phase Kinetex EVO C18 column (250 × 4.6 mm). The photometric assay was carried out in 96-well white assay plates (Costar 655094, Grainer Bio-One, Kremsmünster, Austria) using the Synergy™ Mx Multi-Mode microplate reader (Bio Tek Instruments, Winooski/Vermont, VT, USA). Expressions strain of dCK was kindly provided by BioNukleo GmbH (Berlin, Germany). Enzyme concentrations were determined with a NanoDrop 1000 (Thermo Fisher Scientific, Wilmington, DE, USA).

### 4.2. Expression of HsdCK

Human dCK was expressed in *E. coli* BL21 using EnPresso B medium (Enpresso, Berlin, Germany) based on the manufacturer´s recommendations in a volume of 50 mL. A preculture in LB medium was inoculated from one glycerol stock of *Hs*dCK. All three glycerol stocks used in this study were

from the same working cell bank, which is based on one single colony. To induce protein expression, an IPTG concentration of 50 μM was applied. Cells were harvested by centrifugation. Cells from 4 parallel expression cultures were harvested to one reaction tube. About 2 g of cell pellet$_{wet\ weight}$ were obtained from an expression culture in 50 mL medium.

### 4.3. Immobilization of HsdCK

Three independent enzyme immobilizations denoted as immobilization experiments 1, 2 and 3 were performed with five replicas for each experiment (Figure S1). For each immobilization experiment, *Hs*dCK was freshly expressed in EnPresso B medium (4x 50 mL) from an independent glycerol stock for each experiment.

For one immobilization experiment, 7.5 g of cell pellet were dissolved in 22.5 mL of lysis buffer (50 mM sodium phosphate (pH 8.0), 300 mM NaCl, 10 mM imidazole, 1 mg mL$^{-1}$ lysozyme, 0.6 μg mL$^{-1}$ DNase, 1 mM MgCl$_2$ and 0.1 mg mL$^{-1}$ PMSF). After the enzymatic lysis of 30 min at 30 °C and 300 rpm, the extract was lysed by sonification on ice for 5 min with 30% power input and 30 s on/off intervals. The extract was centrifuged afterwards at 10000 x g for 20 min, and the lysate containing the soluble fraction of the enzyme preparation was collected.

To start the immobilization procedure, Ni-NTA superflow was activated by washing 100 μL of Ni-NTA resin with 1 mL of lysis buffer. The washing step was repeated three times in total. Following the matrix activation, one fifth (4.5 mL) of the lysate containing the soluble protein was loaded to the equilibrated Ni-NTA matrix in 3 successive steps (each 1.5 mL). For each step, the Ni-NTA resin and the lysate were incubated at room temperature for 60 mins, followed by brief centrifugation to remove the supernatant. Following the loading, the matrix was washed three times, each with 1 mL washing buffer (50 mM sodium phosphate (pH 8.0), 300 mM NaCl and 20 mM imidazole) to remove any non-specifically bound enzyme.

Two out of five replicates for each immobilization experiment were used for protein elution. The bound protein was eluted three times (E1, E2, E3), each with 100 μl of elution buffer (50 mM sodium phosphate (pH 8.0), 300 mM NaCl and 250 mM imidazole). The volume of the elution buffer was determined based on preliminary experiments (Figure S2).

Human dCK was dialyzed against 50 mM Tris-HCl (pH 7.4). The protein purity was checked by SDS-PAGE according to a standard protocol [34], and enzyme concentrations were determined with NanoDrop 1000 (Thermo Fisher Scientific, Wilmington, DE, USA) at a wavelength of 280 nm. As negative controls, elution buffer and binding buffer were used. Protein concentrations were determined using an extinction coefficient value calculated based on the amino acid composition of *Hs*dCK ($\varepsilon_{280\ nm}$ = 56380 M$^{-1}$ cm$^{-1}$). To guarantee that imidazole is not interfering with measurements at 280 nm, the absorbance of the applied solutions was analyzed by NanoDrop using deionized water as blank (Figure S5). Imidazole was showing no absorbance in the concentrations applied in this study. Standard deviations were calculated from two replicas of the respective immobilization experiment.

### 4.4. Standard Enzymatic Assay

The following conditions were set as the standard for the whole study. In a final volume of 1 mL a reaction mixture containing 0.333 mM nucleoside (substrate), 0.4 mM phosphate donor (GTP), 0.01 mg mL$^{-1}$ *Hs*dCK and 10 mM MgCl$_2$ in 2 mM potassium phosphate buffer (IUPAC name: Potassium dihydrogen phosphate; dipotassium hydrogen phosphate) (pH 7) was incubated at 37 °C with continuous horizontal shaking at 200 rpm. The reactions were started by the enzyme addition (immobilized or soluble). At defined time points, samples were taken for further analysis (photometric assay or HPLC). For the photometric assay, samples were stored on ice until further usage. For HPLC analysis, samples were transferred to an equal volume of methanol. After centrifugation at 215,000× *g* for 10 min, the supernatant was analyzed by HPLC.

### 4.5. Screening of the Substrate Spectrum of Soluble and Immobilized HsdCK

The screening assay was performed to get a qualitative impression on the substrate spectrum of heterologously expressed *Hs*dCK in the soluble and immobilized form. For the screening, seven natural and 11 modified nucleoside analogues were used as substrates in a reaction volume of 0.5 mL. Before the reaction was started, the three replicas of immobilization experiment 3 were pooled. The reactions were performed in duplicates. The negative control was prepared and incubated in the same manner without the addition of the enzyme to ensure the substrates' stability under the reaction conditions. The phosphorylation of the nucleosides was determined after 20 h incubation to guarantee that product formation is observed with also poor substrates. The photometric assay was applied as described below to quantify product formation based on the depletion of GTP. Standard deviations were calculated from two in independent experiments.

### 4.6. Determination of Specific Activities for NMP Analogues

NMP analogues of cladribine, clofarabine and fludarabine were synthesized under standard conditions in a total volume of 1 mL for 1 h using soluble and immobilized *Hs*dCK. Regular samples were taken to observe a linear depletion of the phosphate donor GTP. Samples were analyzed using the photometric assay as described below. Standard deviations were calculated from two (soluble enzyme) and three (immobilized enzyme) in independent experiments.

The retained activity was calculated from the quotient of the specific activity of the immobilized enzyme to the free enzyme (Equation (1)).

$$\text{Retained activity } [\%] = \frac{\text{specific activity of the immobilized enzyme } [\text{U/mg}]}{\text{specific activity of the soluble enzyme } [\text{U/mg}]} \times 100\% \qquad (1)$$

The conversion percentages were calculated based on GTP depletion. One unit (U) of enzyme activity was defined as the amount of enzyme required for the conversion of 1 $\mu$mol substrate per minute under the given reaction conditions.

### 4.7. Optimization of Reaction Conditions

To investigate parameters influencing the reaction yield, different conditions, including the phosphate donor and its ratio to the substrate and the $MgCl_2$ were examined (Table 2). All reactions were performed with fludarabine in a 1 mL volume for 1 h. Before the reaction was started, the three replicas of immobilization experiment 3 were pooled. The reactions were performed in duplicates.

To evaluate the efficiency of different phosphate donors, ATP, GTP and UTP were used alternatively under standard reaction conditions.

The ratio of phosphate donor to the substrate was as well tested. Ratios of 1:1, 1:2 and 1:4 fludarabine to GTP were used. While maintaining all reaction conditions as the standard reaction, fludarabine concentration was decreased from 0.4 to 0.2 and 0.1 mM.

The effect of different $MgCl_2$ concentration was investigated by applying different ratios of phosphate donor (GTP) to $MgCl_2$. $MgCl_2$ with concentrations of 0.4 mM, 1 mM and 2 mM were used in the standard reaction.

For all the reactions, samples were taken over 1 h with 10 minutes intervals and analyzed using the photometric assay as described below. Standard deviations of the reactions catalyzed by the soluble enzyme were calculated from two independent experiments.

### 4.8. Increasing the Substrate Concentration for Fludarabine-5´-Monophosphate Synthesis

The concentrations of fludarabine (0.333 mM), GTP (0.4 mM) and *Hs*dCK (0.01 mg mL$^{-1}$) were increased by a factor of 3, 19.5, or 36 times, while maintaining the ratio of fludarabine to GTP (1:1.2) and that of fludarabine to *Hs*dCK (1:0.03). 1 mM, 6.5 mM and 12 mM fludarabine were used with 1.2 mM, 7.8 mM and 14.4 mM GTP, respectively, in the presence of 0.03 mg mL$^{-1}$, 0.195 mg mL$^{-1}$ and

0.36 mg mL$^{-1}$ *Hs*dCK, respectively. Reactions were performed in a final volume of 0.1 mL in 2 mM potassium phosphate buffer (pH 7) containing 10 mM MgCl$_2$. After 18 h, reactions were analyzed by HPLC as described below. Standard deviations were calculated from two independent experiments.

### 4.9. Reusability of the Immobilized Enzyme

1 mL standard reaction using fludarabine as substrate and immobilized *Hs*dCK was incubated at 37 °C. After 3 h, the enzyme was collected by centrifugation at 500 rpm for 5 min and re-used. Immobilized *Hs*dCK was used for three consecutive reactions denoted by R1, R2 and R3. Samples were analyzed using HPLC as described below. Standard deviations were calculated from triplicate reactions.

The relative retained activity used to study the reusability of *Hs*dCK was calculated based on Equation (2).

$$\text{Relative retained activity } [\%] = \frac{\% \text{ of formation of fludarabine} - 5' - \text{monophosphate in consecutive reactions}}{\text{mean value for \% of formation of fludarabine} - 5' - \text{monophosphate in the first rection}} \times 100\% \tag{2}$$

### 4.10. Photometric Assay

The formation of NMPs was detected photometrically using the Kinase-Glo® Max assay (Promega, Madison/Wisconsin, USA). The assay is used to quantify product formation based on the depletion of GTP. Samples were taken from the reaction at different time points and stored on ice until further analysis. In a 96-well plate, 10 μL of Kinase-Glo® reagent was added to 10 μL of each sample and 80 μL deionized water. After 10 min of incubation in the dark, luminescence was measured for 1s over an extended dynamic range. The conversion percentages of nucleosides to NMPs were calculated based on the nucleotide (phosphate donor) depletion.

### 4.11. High Performance Liquid Chromatography (HPLC)

Nucleoside/nucleotide depletion and nucleotide formation were validated by HPLC (λ = 260 nm) using a reversed-phase Kinetex EVO C18 column at 34 °C with a flow rate of 1 mL min$^{-1}$. Substrates and products were identified by comparing their retention times with those of authentic standards. Quantification was performed using a 6-step calibration in the range from 0 mM to 1 mM over the peak area.

Gradual elution was performed using eluent A (8 mM tetrabutylammonium bisulfate in 100 mM potassium phosphate buffer, pH 5.4) and eluent B (70% eluent A and 30% methanol). The following gradient was used: After 10 min of 80% A and 20% B, a gradual change to 40% A and 60% B over 10 min, then to 38% A and 62% B over 12 min. The initial condition (80% A and 20% B) was restored within 1 min and maintained for 3 min.

The following retention times were observed: GMP, 4.4 min; 2F-Ade, 6.5 min; GDP, 9.3 min; 2F-AraA, 10.6 min; 2F-Ara-AMP, 19.4 min; GTP, 22.1 min. Yields were determined according to Equation (3). Standard deviations were calculated from three independent experiments unless otherwise specified.

$$\text{Reaction yield } [\%] = \frac{\text{conc.of the formed nucleotide } [\text{mM}]}{\text{conc.of the formed nucleotide } [\text{mM}] + \text{conc of nucleoside } [\text{mM}]} \times 100\%. \tag{3}$$

## 5. Conclusions

Like its natural variant heterologously expressed *Hs*dCK accepts a wide variety of deoxyribonucleosides and their analogues as substrates. The immobilization onto Ni-NTA superflow is a suitable option to enhance the purification process by allowing the buffer system to be changed easily and allowing the reusability of the enzyme. Although activities of the immobilized *Hs*dCK were lower

compared to the soluble enzyme at low enzyme concentration, activity significantly increased with increasing substrate concentration. Hence, this study shows that *Hs*dCK is a valuable biocatalyst for the synthesis of a wide range of high-added value NMP analogues, including fludarabine-5′-monophosphate, cladribine-5′-monophosphate and clofarabine-5′-monophosphate under industrially-relevant conditions. Furthermore, immobilized *Hs*dCK can be used to study the efficient phosphorylation of new drug candidates in in vitro screening assays.

**Supplementary Materials:** The following are available online at http://www.mdpi.com/2073-4344/9/12/997/s1.

**Author Contributions:** A.W. (Anke Wagner) and K.F.H. conceived and designed the experiments. K.F.H. performed and A.W. (Albane Wetterwald) assisted with the experiments. A.W. (Anke Wagner), S.K., K.F.H. and P.N. wrote and revised the paper.

**Funding:** This research was funded by the Deutsche Forschungsgemeinschaft (DFG, German Research Foundation), grant number 392246628.

**Acknowledgments:** K.F.H. is funded by the Deutsche Forschungsgemeinschaft (DFG, German Research Foundation). The authors are greatly thankful to Erik Wade for proofreading the manuscript and critical comments. We are very grateful that a grant was provided from the publication fund of the TU Berlin.

**Conflicts of Interest:** A.W. (Anke Wagner) is CEO and P.N. is a member of the advisory board of BioNukleo GmbH. S.K. is a scientific researcher, and A.W. (Albane Wetterwald) was a student worker at BioNukleo GmbH. The authors have no other relevant affiliations or financial interests in or financial conflicts with the subject matter or materials discussed in the manuscript apart from those disclosed. No potential conflict of interest is known for the other authors.

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
