# Peer review of "Human Deoxycytidine Kinase Is a Valuable Biocatalyst for the Synthesis of Nucleotide Analogues"

_catalysts, doi:10.3390/catal9120997_

Round 1

Reviewer 1 Report

This is a report about the biocatalytic synthesis of modified nucleotides by employing an immobilized human deoxycytidine kinase. Human dCK was immobilized onto Ni-NTA superflow and a screening of its substrate spectrum was performed. After optimization of the reaction conditions, the immobilized biocatalyst was used for the synthesis of fludarabine monophosphate in an up-scaling experiment (substrate concentrations ranging from 1 mM to 12 mM). In summary, authors have developed a greener synthetic methodology for the synthesis of modified nucleotides. The report is well written and organized, and it presents some interesting results. However, I recommend rejecting the manuscript since a lot of information and detailed explanations are missing. Accordingly, some of the major issues found in the manuscript are listed below.

Immobilization of hdCK

Overall, the immobilization process must be explained in greater detail.

First of all, protein loading values must be provided. In figure 1 the amount of enzyme immobilized onto the support is shown, though it is not related with the amount of support employed. In spite of knowing that 100 µL of support were used, without the concentration value it is not possible to know the amount of support needed for the immobilization process. In this sense, values about the amount of immobilized enzyme per gram of support must be provided in the results section. Moreover, different immobilization conditions can be explored in order to improve the immobilization yield. The process employed for quantifying the bound enzyme must be further explained. In addition, some related points must be clarified: Did the authors use imidazole containing buffers during the immobilization and elution process? Imidazole absorbs UV radiation at 280 nm depending on its source and purity, so Nanodrop 1000 should not be used for accurate quantification. In such cases, Bradford protein assay is recommended for protein quantification. The purity of the eluted enzyme was checked with SDS PAGE, but this topic is not further mentioned in the manuscript. First of all, hdCK was purified directly from cell extracts by immobilization onto Ni-NTA support, so information about purity is required in order to evaluate the efficiency of the purification process. Secondly, since authors evaluated the enzymatic activity for synthesis of modified nucleotides, information about the purity of the immobilized biocatalyst must be provided. The immobilized biocatalyst must be better characterized. Enzymatic activity values from both immobilized and free enzyme must be compared and resulting retained activity rates must be shown in the results section. In relation to points 1 and 3, authors must provide a table showing the amount of enzyme immobilized per gram of support and the retained activity rates when compared with the free enzyme. Finally, for better characterization, values of enzymatic activity per gram of support must be also shown in the table.

Enzymatic synthesis of NMP analogues and optimization of the reaction conditions

Conversion percentages are not accurate units when talking about the synthesis of NMP analogues, so substrate specificity of hdCK must be explained in terms of specific activity. In this sense, when the synthesis of NMP analogues is addressed in the manuscript, specific activity values must be provided instead of formation percentages. Finally, specific activity values must be also shown in both table 1 and table 2. Screening of the substrate spectrum was performed by evaluating the formation of NMP and analogues after 20 h of reaction. Before incubating the enzyme for such long periods, the thermal stability of both immobilized and free enzyme should be checked. Consequently, if the enzyme does not withstand the latter conditions and becomes denatured, it does not make sense to carry out such long synthesis processes.

Upsclaing of fludarabine monophosphate synthesis

Fludarabine monophosphate formation was expressed by means of conversion rate. Once again, percentages of formation are not valid units for evaluating the product formation. In this respect, applying these percentages, authors must show the amount of product generated after the enzymatic reaction. Thus, figure 3 must be corrected including concentrations of the generated product. Another alternative would be including a table showing such concentration values for each substrate concentration tested.  As in the previous section, screening of the substrate spectrum, fludarabine monophosphate formation was carried out by a long enzymatic process (18 h). Before incubating the enzyme for such long periods, the thermal stability of both immobilized and free enzyme must be checked, and results must be provided. In this sense, it would be interesting to compare the obtained results for both immobilized and free enzyme to confirm that enzyme immobilization improves the thermal stability.

Reusability of the immobilized hdCK

The immobilized enzyme was used for two consecutive reactions without any significant loss of activity, while 95% of the enzyme activity was lost afterwards. Before using the immobilized biocatalyst in three consecutive three-hour reaction cycles, its thermostability must be thoroughly evaluated. In this sense, if the immobilized enzyme does not retain the activity for more than 6h at 37 ° C, it does not make much sense to carry out a third reaction cycle. To better show the difference in activity for each reaction cycle, relative activity rates must be provided. In other words, enzymatic activity value in the first cycle must be considered as the maximum activity that can be achieved (100% of relative activity), so the activity values for the rest of cycles must be referred to the latter activity rate.  

Discussion

Unfortunately, authors did not make a real discussion of the results. Overall in this section, the results obtained are just mentioned, but not really discussed.

First of all, substrate spectrum of free hdCK was compared to previously reported works, but no further explanation is presented about the reason for this substrate specificity. The differences among the immobilized catalyst and the free enzyme are not discussed in depth. In this sense, authors must mention the advantages obtained due to the immobilization of the enzyme. Finally, immobilized hdCK is compared to immobilized DmdNK and Dd Once again, specific activity values instead of conversion percentages must be compared to get relevant information. Authors mentioned that changing the immobilization support and protocol might lead to increased activity, however, the reason for the decrease in activity when immobilized must be better explained.

Materials and methods

Despite having followed the commercial protocol, the immobilization process must be explained in more detail. Authors have provided the retention times of the studied compounds. However, how did the authors perform the identification and quantification of the resulting products? Was it done in relation to external standards? Some information is required to clarify this point.

Author Response

This is a report about the biocatalytic synthesis of modified nucleotides by employing an immobilized human deoxycytidine kinase. Human dCK was immobilized onto Ni-NTA superflow and a screening of its substrate spectrum was performed. After optimization of the reaction conditions, the immobilized biocatalyst was used for the synthesis of fludarabine monophosphate in an up-scaling experiment (substrate concentrations ranging from 1 mM to 12 mM). In summary, authors have developed a greener synthetic methodology for the synthesis of modified nucleotides. The report is well written and organized, and it presents some interesting results. However, I recommend rejecting the manuscript since a lot of information and detailed explanations are missing. Accordingly, some of the major issues found in the manuscript are listed below.

Thank you for the very helpful comments. Further Information was added to the manuscript based on the suggestions.

Immobilization of hdCK

Overall, the immobilization process must be explained in greater detail.

More detailed information was added to the manuscript.

First of all, protein loading values must be provided. In figure 1 the amount of enzyme immobilized onto the support is shown, though it is not related with the amount of support employed. In spite of knowing that 100 µL of support were used, without the concentration value it is not possible to know the amount of support needed for the immobilization process.

More detailed information was added to the materials and methods section and Figure 1.

In this sense, values about the amount of immobilized enzyme per gram of support must be provided in the results section.

The supplier of Ni-NTA is only providing volume information for the support (Qiagen, Germany). This is probably due to the fact that Ni-NTA sepharose is not a solid support but has a gel-like appearance. The binding capacity is given as maximum 50 mg per 1 ml of support. However, it is also described that the yield depends on the binding capacity of the specific protein. With our preparations we observed a binding of maximum 1.2 mg of hdCK to 100 µl of Ni-NTA superflow material. More detailed information are also given in the results section.

Moreover, different immobilization conditions can be explored in order to improve the immobilization yield.

This is a valuable hint; however, it is not possible in the short time available to optimize the immobilization conditions. Purified human dCK was used extensively before in our laboratories and the purification procedure was optimized in preliminary experiments. The amount of loaded cell extract and the washing procedure was optimized to reach sufficient protein yield and purity. These conditions optimized in preliminary experiments were applied for the immobilization strategy used in this study. Further information regarding these experiments were added to the manuscript.

Based on the results obtained, we tend to use a completely different strategy for future experiments including ionic interaction and cross-linking.

The process employed for quantifying the bound enzyme must be further explained.

Further information has been added to the materials and methods section.

In addition, some related points must be clarified: Did the authors use imidazole containing buffers during the immobilization and elution process? Imidazole absorbs UV radiation at 280 nm depending on its source and purity, so Nanodrop 1000 should not be used for accurate quantification. In such cases, Bradford protein assay is recommended for protein quantification.

Thank you very much for this hint. The imidazole applied in this study was tested for absorbance at 280 nm. No absorbance at 280 nm was observed with imidazole concentrations used in the elution buffer (250 mM). After dialysis, imidazole concentration should even be strongly reduced.

The purity of the eluted enzyme was checked with SDS PAGE, but this topic is not further mentioned in the manuscript. First of all, hdCK was purified directly from cell extracts by immobilization onto Ni-NTA support, so information about purity is required in order to evaluate the efficiency of the purification process. Secondly, since authors evaluated the enzymatic activity for synthesis of modified nucleotides, information about the purity of the immobilized biocatalyst must be provided.

An example for the purification of human dCK has been inserted in Figure 1. Similar results were achieved in all purification approaches. In addition, descriptive text was inserted in the results section.

The immobilized biocatalyst must be better characterized. Enzymatic activity values from both immobilized and free enzyme must be compared and resulting retained activity rates must be shown in the results section. In relation to points 1 and 3, authors must provide a table showing the amount of enzyme immobilized per gram of support and the retained activity rates when compared with the free enzyme.

Values for retained activity were added to the text; lines 216-220 and Table 1. Retained activity was calculated based on specific activities for both soluble and immobilized biocatalyst. The amount of protein bound to the matrix is shown in Figure 1C.

Finally, for better characterization, values of enzymatic activity per gram of support must be also shown in the table.

Respective information was added to Supplementary table 1.

Enzymatic synthesis of NMP analogues and optimization of the reaction conditions

Conversion percentages are not accurate units when talking about the synthesis of NMP analogues, so substrate specificity of hdCK must be explained in terms of specific activity. In this sense, when the synthesis of NMP analogues is addressed in the manuscript, specific activity values must be provided instead of formation percentages.

We agree, additional information is provided, when specific activities are given. Therefore, specific activities were determined for a set of substrates of human dCK: cladribine, fludarabine and clofarabine (Table 1).

From a synthesis perspective, an additional important information is the formation percentage as different reaction conditions might influence final product yields. Additionally, conversion percentages were also shown in publications to describe dNKs from different sources. Serra et al., 2017 immobilization of deoxyadenosine kinase from Dictyostelium discoideum (DddAK) and its application in the 5’-Phosphorylation of arabinosyladenine and arabinosyl-2-fluoroadenine. Serra et al. 2014 also immobilized Drosophila melanogaster deoxyribonucleoside kinase (DmdNK) as a high performing biocatalyst for the synthesis of purine arabinonucleotides.

Finally, specific activity values must be also shown in both table 1 and table 2.

Specific activities are shown in Table 1. For further approaches conversion percentage was the most important factor as we were aiming to produce as much product as possible. Specific activity itself is not telling about the percentage of product formed in the reaction.

Screening of the substrate spectrum was performed by evaluating the formation of NMP and analogues after 20 h of reaction. Before incubating the enzyme for such long periods, the thermal stability of both immobilized and free enzyme should be checked. Consequently, if the enzyme does not withstand the latter conditions and becomes denatured, it does not make sense to carry out such long synthesis processes.

The stability of the enzyme was studied. An additional figure was added to the Supplementary material. Further information was also added to the text.

We also tested for the stability of the phosphate donor GTP and we observed that it was stable of a period of 24h.

Upsclaing of fludarabine monophosphate synthesis

Fludarabine monophosphate formation was expressed by means of conversion rate. Once again, percentages of formation are not valid units for evaluating the product formation. In this respect, applying these percentages, authors must show the amount of product generated after the enzymatic reaction. Thus, figure 3 must be corrected including concentrations of the generated product. Another alternative would be including a table showing such concentration values for each substrate concentration tested.  

A Table including the concentrations were added to the Supplementary material (Supplementary table 2). Results were also described in more detail in the manuscript.

As in the previous section, screening of the substrate spectrum, fludarabine monophosphate formation was carried out by a long enzymatic process (18 h). Before incubating the enzyme for such long periods, the thermal stability of both immobilized and free enzyme must be checked, and results must be provided. In this sense, it would be interesting to compare the obtained results for both immobilized and free enzyme to confirm that enzyme immobilization improves the thermal stability.

We agree that it would be beneficial to increase the thermal stability by immobilization. However, using Ni-NTA as support the matrix itself is a critical factor. Using Ni-NTA sepharose the reaction temperature cannot be increased to values above 40°C. Even with short reaction times, the matrix is melting.

As stated before, we were addressing the stability of human dCK at 37°C. Studies were performed (Supplementary figure 2) and it was shown that both free and immobilized human dCK are stable at 37°C over a period of 24h.

Reusability of the immobilized hdCK

The immobilized enzyme was used for two consecutive reactions without any significant loss of activity, while 95% of the enzyme activity was lost afterwards. Before using the immobilized biocatalyst in three consecutive three-hour reaction cycles, its thermostability must be thoroughly evaluated. In this sense, if the immobilized enzyme does not retain the activity for more than 6h at 37 ° C, it does not make much sense to carry out a third reaction cycle.

The stability was evaluated as mentioned above and shown in Supplementary figure 2 and the enzyme was stable over a period of 24 hours. A challenge of using Ni-NTA sepharose as support was the binding of the enzyme to the matrix. Therefore, an explanation for losses in activity might be a detachment of the biocatalyst from the matrix during the washing steps and the incubation (including pipetting and mixing). Hence, with increasing steps and incubation time enzyme is detached and activity is reduced in the next reaction. The observation was in agreement with previous observations where protein leach from the matrix was described with time, when non-covalent immobilization was applied (Homaei et al., 2013).

Additional text was added for clarification.

To better show the difference in activity for each reaction cycle, relative activity rates must be provided. In other words, enzymatic activity value in the first cycle must be considered as the maximum activity that can be achieved (100% of relative activity), so the activity values for the rest of cycles must be referred to the latter activity rate.

An additional figure was added to the supplementary material to show relative activity rates (Supplementary figure 3).

Discussion

Unfortunately, authors did not make a real discussion of the results. Overall in this section, the results obtained are just mentioned, but not really discussed.

Many thanks for the very detailed and helpful hints for the discussion. The discussion was significantly changed to make the important points clearer and to give explanations for the observed results.

First of all, substrate spectrum of free hdCK was compared to previously reported works, but no further explanation is presented about the reason for this substrate specificity. The differences among the immobilized catalyst and the free enzyme are not discussed in depth. In this sense, authors must mention the advantages obtained due to the immobilization of the enzyme.

Further information was added to the discussion to better characterize soluble and immobilized enzyme. It was highlighted that immobilized enzyme is advantageous when using higher substrate concentrations.

Finally, immobilized hdCK is compared to immobilized DmdNK and Dd Once again, specific activity values instead of conversion percentages must be compared to get relevant information.

Values for specific activity was added to the discussion.

Authors mentioned that changing the immobilization support and protocol might lead to increased activity, however, the reason for the decrease in activity when immobilized must be better explained.

An explanation for the loss in activity was added to the discussion section.

Materials and methods

Despite having followed the commercial protocol, the immobilization process must be explained in more detail.

The immobilization protocol is now described in detail in the materials and methods section and in Figure 1.

Authors have provided the retention times of the studied compounds. However, how did the authors perform the identification and quantification of the resulting products? Was it done in relation to external standards? Some information is required to clarify this point.

Further information was added to the materials and methods section for clarification.

Reviewer 2 Report

Human deoxycytidine kinase is a valuable biocatalyst 2 for the synthesis of nucleotide analogues

The above study is very well executed in studying the kinetics of human deoxycytidine kinase. Kinases utilizing nucleoside towards biosynthesis of nucleotides is a very important enzymatic reaction in developing drugs as antiviral agents.

With growing evidence of resistance, assays help to interrogate the analogs at early stage and this expression system and assay will be useful.

One concern for the reviewer was the AUC reports from the HPLC towards quantification of nucleotides.

Author Response

Human deoxycytidine kinase is a valuable biocatalyst 2 for the synthesis of nucleotide analogues

The above study is very well executed in studying the kinetics of human deoxycytidine kinase. Kinases utilizing nucleoside towards biosynthesis of nucleotides is a very important enzymatic reaction in developing drugs as antiviral agents.

With growing evidence of resistance, assays help to interrogate the analogs at early stage and this expression system and assay will be useful.

Thank you for your comments on our manuscript.

One concern for the reviewer was the AUC reports from the HPLC towards quantification of nucleotides.

Additional information was added to the materials and methods section to clarify the identification and quantification procedures of the nucleotides.

Reviewer 3 Report

The manuscript by Hellendal and colleagues evaluates the potential of human deoxycytidine kinase as a suitable catalyst for the synthesis of ribonucleoside monophosphates.

This is an interesting work as it offers a promising approach for the synthesis of valuable compounds.

The manuscript should be suitable for publication but only after it has been revised to address the following points.

The manuscript is difficult to assess in the present form as it lacks many experimental details. Moreover, other experimental data must be more clearly presented.

Line 78: “Human dCK was immobilized onto Ni-NTA superflow using the standard protocol of the manufacturer (Figure 1A).”How was conducted the immobilization? What were the volume of the cell extract and its protein content? Was the recombinant enzyme efficiently produced in E. coli? These data are important to evaluate the efficiency of the coupling between the enzyme and the Ni-NTA matrix. SDS-PAGE analysis of the human enzyme before and after purification should be shown.

As a consequence, it is difficult the apprehend data presented in Figure 1B. What is the volume of the elution fractions? What is the explanation of the low yield of the third experiment? What is the purity of the enzyme in each experiment? Details regarding the error bars must be provided (in Figure 1B and in the other Figures of the manuscript).

In the legend of Figure 2, there is no mention of the presence of MgCl2. However, in the Material and Methods section, the standard conditions are described as containing 10 mM MgCl2. Thus, if added for the experiments, the presence of MgCl2 should be indicated in the legend.

Regarding the standard conditions, it is mentioned (line 281) an enzyme concentration of 0.01mg/ml. However, the legend of Figure reports 0.01 mg of enzyme in a final volume of 0.5 ml (thus a concentration of 0.02 mg/ml). Are the conditions described in Figure 2 different than standard conditions?

Line 120: “When analyzing specific activities and percentage of conversion determined for cladribine, fludarabine and clofarabine, higher values were observed for soluble enzyme compared with immobilized enzyme with a few exceptions (Table 1)”. Please explain in detail what these exceptions are.

Line 122: “The impact…” What exactly was meant? The specific activity or the conversion?

In Figure 2B, after a reaction time of 20 hours with an enzyme concentration of 0.02 mg/ml (see above), cladribine (2C-dAdo) conversion is reported to be approximately 40% for the immobilized enzyme. In contrast, results in Table 1 indicate that after a reaction time of only 60 min and an enzyme concentration of 0.01 mg/ml, the conversion is significantly higher: 52% (in the absence of DTT). Please comment.

Lines 135-137: “Using the standard reaction conditions, product yields of 21% and 54% were observed using immobilized and soluble hdCK as biocatalyst, respectively, after a reaction time of one hour (Table 2)”. The experimental conditions used for experiments reported in Table 2 seem to be identical to those related to the experiments described in Table 1. However, the error values indicated in brackets (SD?) in Table 1 and Table 2 differ by an order of magnitude for Fludarabine (6 and 7% in Table 1 versus 0.6 and 0.4%). Please comment.

Line 189: How many replicates were used for the experiments of Figure 3? Please indicate what the bars represent. What were the experimental conditions?

Line 185: “100% formation using the standard conditions”. Do the authors refer to data presented in Figure 2B? Please indicate. If this is the case, the authors compare product formation in the presence of 0.02 mg/ml of enzyme for a period of 20 hours (Figure 2B) and product formation in the presence of 0.01 mg/ml of enzyme for a period of 18 hours (Figure 4). Please comment and indicate the experimental conditions used for Figure 4. How do the authors explain the increase of product formation with immobilized enzyme? This is an important point since it is stated that biocatalyst immobilization is an advantage.

Line 213-215: “While for the native enzyme UTP was a better phosphate donor than ATP [21], the heterologously expressed variant prefers GTP and ATP over UTP (Table 2).” It cannot be stated that the studied enzyme prefers GTP over UTP in the absence of a detailed kinetic study. The lower conversion observed in the present study can be due to substrate inhibition, as described by TL Hugues et al.: “Kinetic Analysis of Human Deoxycytidine Kinase with the True Phosphate Donor Uridine Triphosphate” in Biochemistry 1997, 36, 7540-7547. Please modify the text accordingly.

In the discussion section, the authors report from the literature higher conversion for immobilized enzymes from Drosophila melanogaster or Dictyostelium discoideum. The authors should discuss this further. Why would be the benefit from using the human enzyme if it is less active? The authors do not discuss the cause of the lower conversion of the immobilized enzyme. Does the Ni-NTA binding alter the access of the substrate(s) to the active site? What could be the alternative to the Ni-NTA binding? These important aspects should be discussed and would benefit the manuscript.

Minor comments:

Line 94: change “thymdine” to “thymidine”

Lines 104-105: “the addition of dichlorodiphenyltrichloroethane (DTT)”. Dichlorodiphenyltrichloroethane is usually abbreviated in DDT, which is not a reducing agent ( line 105 :“reducing agents like DTT”). Please explain what DTT is the present study (dithiothreitol?).

Line 117: “2CdA: cladribine”. In Figure 2B, cladribine seems to appear as “2C-dAdo”. Please modify.

Line 182: “(Figure 4, 1B)”. Apparently, there is no panel 1B in Figure 4. Please modify.

Author Response

The manuscript by Hellendal and colleagues evaluates the potential of human deoxycytidine kinase as a suitable catalyst for the synthesis of ribonucleoside monophosphates.

This is an interesting work as it offers a promising approach for the synthesis of valuable compounds.

The manuscript should be suitable for publication but only after it has been revised to address the following points.

The manuscript is difficult to assess in the present form as it lacks many experimental details. Moreover, other experimental data must be more clearly presented.

Thank you for the helpful comments. They helped to improve the manuscript. Changes were made as suggested.

Line 78: “Human dCK was immobilized onto Ni-NTA superflow using the standard protocol of the manufacturer (Figure 1A).”How was conducted the immobilization? What were the volume of the cell extract and its protein content? Was the recombinant enzyme efficiently produced in E. coli? These data are important to evaluate the efficiency of the coupling between the enzyme and the Ni-NTA matrix. SDS-PAGE analysis of the human enzyme before and after purification should be shown.

A more detailed explanation of the immobilization procedure was added to the materials and methods section and the results section next to information to the volume of cell extract, and the protein content of the cell extract.

A picture of an SDS-PAGE was added. Expression in BL21 is shown in the soluble fraction of the cell extract. By using 1.5 g of cell pellet per 100 µl of Ni-NTA matrix for the immobilization or purification procedure target protein was efficiently bound to the matrix. Only few impurities were observed in the elution fractions. The respective results were described in more detail.

As a consequence, it is difficult the apprehend data presented in Figure 1B. What is the volume of the elution fractions? What is the explanation of the low yield of the third experiment? What is the purity of the enzyme in each experiment? Details regarding the error bars must be provided (in Figure 1B and in the other Figures of the manuscript).

Additional information has been added to the manuscript. Information to error bars were added to all tables and figures of the manuscript.

In the legend of Figure 2, there is no mention of the presence of MgCl2. However, in the Material and Methods section, the standard conditions are described as containing 10 mM MgCl2. Thus, if added for the experiments, the presence of MgCl2 should be indicated in the legend.

The applied amount of MgCl2 was added to the legend of Figure 2.

Regarding the standard conditions, it is mentioned (line 281) an enzyme concentration of 0.01mg/ml. However, the legend of Figure reports 0.01 mg of enzyme in a final volume of 0.5 ml (thus a concentration of 0.02 mg/ml). Are the conditions described in Figure 2 different than standard conditions?

Unfortunately, it was a spelling mistake. In the screening assay an enzyme concentration of 0.01 mg/ml was applied which corresponds to 0.005 mg in a reaction volume of 0.5 ml. The spelling error was corrected.

Line 120: “When analyzing specific activities and percentage of conversion determined for cladribine, fludarabine and clofarabine, higher values were observed for soluble enzyme compared with immobilized enzyme with a few exceptions (Table 1)”. Please explain in detail what these exceptions are.

A sentence was added to explain the exceptions: “Only for reactions with cladribine and clofarabine in the presence of 5 mM DTT specific activity was slightly higher for the immobilized proteins.”

Line 122: “The impact…” What exactly was meant? The specific activity or the conversion?

The sentence was re-phrased: “The impact of immobilization on the specific activity and the percentage of conversion was more pronounced for cladribine and fludarabine, while differences were minor for clofarabine.”

In Figure 2B, after a reaction time of 20 hours with an enzyme concentration of 0.02 mg/ml (see above), cladribine (2C-dAdo) conversion is reported to be approximately 40% for the immobilized enzyme. In contrast, results in Table 1 indicate that after a reaction time of only 60 min and an enzyme concentration of 0.01 mg/ml, the conversion is significantly higher: 52% (in the absence of DTT). Please comment.

In both the screening assay and the experiments performed to determine specific enzyme activities an enzyme concentration of 0.01 mg/ml was used.

Differences in the results shown in Figure 2 and Table 1 might be explained by instabilities of the final products. Longer incubation times might therefore lead to decreased percentages of product.

In general, the screening assay was just used to get a qualitative impression on substrates that were accepted by hdCK. For clarification an additional sentence was added to the manuscript: “The screening assay was only applied as a qualitative measure.”

Lines 135-137: “Using the standard reaction conditions, product yields of 21% and 54% were observed using immobilized and soluble hdCK as biocatalyst, respectively, after a reaction time of one hour (Table 2)”. The experimental conditions used for experiments reported in Table 2 seem to be identical to those related to the experiments described in Table 1. However, the error values indicated in brackets (SD?) in Table 1 and Table 2 differ by an order of magnitude for Fludarabine (6 and 7% in Table 1 versus 0.6 and 0.4%). Please comment.

When determining specific activities for cladribine, fludarabine and clofarabine (Table 1) the three replicates of one immobilization experiment were used. Hence, standard deviations were calculated for values from each replicate. For later experiments (including table 2) immobilized biocatalysts from the three replicates were pooled. Percentage of conversion was afterwards determined in duplicates using the pooled biocatalyst. For clarification an additional information was given in the legends of the Tables.

Line 189: How many replicates were used for the experiments of Figure 3? Please indicate what the bars represent. What were the experimental conditions?

Triplicates were used to study the re-usability of immobilized human dCK. Bars represent the formation of fludarabine monophosphate and the consumption of GTP. To make it more clear further information were added to the figure legend including the experimental conditions.

Line 185: “100% formation using the standard conditions”. Do the authors refer to data presented in Figure 2B? Please indicate. If this is the case, the authors compare product formation in the presence of 0.02 mg/ml of enzyme for a period of 20 hours (Figure 2B) and product formation in the presence of 0.01 mg/ml of enzyme for a period of 18 hours (Figure 4). Please comment and indicate the experimental conditions used for Figure 4. How do the authors explain the increase of product formation with immobilized enzyme? This is an important point since it is stated that biocatalyst immobilization is an advantage.

Inhibitory effects caused by the natural substrate deoxycytidine are described in literature. Similar effects could also be observed for modified substrates. It seems that the inhibitory effect is less pronounced for the immobilized biocatalyst. However, further studies are necessary to evaluate this effect. A similar effect was described for DmdNK. While at lower substrate concentration the conversion percentage was >98% under optimized conditions, the soluble enzyme was slower at higher substrate concentration compared to the immobilized biocatalyst. Further information was added to text for clarification.

Line 213-215: “While for the native enzyme UTP was a better phosphate donor than ATP [21], the heterologously expressed variant prefers GTP and ATP over UTP (Table 2).” It cannot be stated that the studied enzyme prefers GTP over UTP in the absence of a detailed kinetic study. The lower conversion observed in the present study can be due to substrate inhibition, as described by TL Hugues et al.: “Kinetic Analysis of Human Deoxycytidine Kinase with the True Phosphate Donor Uridine Triphosphate” in Biochemistry 1997, 36, 7540-7547. Please modify the text accordingly.

Further information was added as suggested.

In the discussion section, the authors report from the literature higher conversion for immobilized enzymes from Drosophila melanogaster or Dictyostelium discoideum. The authors should discuss this further. Why would be the benefit from using the human enzyme if it is less active?

Human dCK is a well-studied enzyme for which more than 40 different substrates were described. It accepts a wide variety of natural and modified pyrimidine und purine nucleosides (both D- and L-form). Hence, human dCK is an interesting biocatalyst for synthesis applications. Much less is known about dNKs form Drosophila melanogaster or Dictyostelium discoideum. Hence, in case both enzymes are not delivering sufficient product yields, its worth having a look to human dCK. Additionally, it some cases it might be beneficial if not all substrates are accepted by a specific enzyme, especially when applying enzyme casdades.

The authors do not discuss the cause of the lower conversion of the immobilized enzyme. Does the Ni-NTA binding alter the access of the substrate(s) to the active site? What could be the alternative to the Ni-NTA binding? These important aspects should be discussed and would benefit the manuscript.

Further information addressing these questions were added.

Minor comments:

Line 94: change “thymdine” to “thymidine”

The spelling error was corrected.

Lines 104-105: “the addition of dichlorodiphenyltrichloroethane (DTT)”. Dichlorodiphenyltrichloroethane is usually abbreviated in DDT, which is not a reducing agent ( line 105 :“reducing agents like DTT”). Please explain what DTT is the present study (dithiothreitol?).

We apologize for the confusion. Dithiothreitol (DTT) was used as a reducing agent because it has been described in the literature that DTT is beneficial for the activity of human dCK.

Line 117: “2CdA: cladribine”. In Figure 2B, cladribine seems to appear as “2C-dAdo”. Please modify.

The figure was changed respectively.

Line 182: “(Figure 4, 1B)”. Apparently, there is no panel 1B in Figure 4. Please modify.

The text was changed respectively.

Round 2

Reviewer 1 Report

Despite authors have answered several suggestions, the manuscript have some grammar mistakes. In addition, the biocatalyst is previously described (it is not a new enzyme), the immobilization process is not optimized, the discussion is poor, the reusability of the biocatalysts is limited to 1 cycle, and the "scale up" is performed at 12 mM substrate concentration in a final volume of 0.1 ml. Because of this, the impact of this research is far from its potential. 

Some minor comments:

Authors should send 2 manuscript versions, the first one with corrections and the second one without any marks, to render the text easier to read. 

ABSTRACT

- Authors should used the term “Ribonucleoside-5’-monophosphates” instead “ribonucleoside monophosphates” in the different sections of the manuscript.

- Authors mentions that “The immobilized enzyme  was successfully re-used”, however after the second reuse the enzyme loss more than 90 % of the activity. In this sense, authors should remove this claim this. In addition, they also need to explain the reason of this activity loss. Error bars for R1 (figure 3) are so high.

- Authors must replace hdCK by HsdCK

RESULTS

-Figure 1. “Immobilization of human dCK to Ni-NTA superflow via ionic interaction” must be replaced by ………………affinity interaction”

-Line 408 “Upsclaing” must be replaced by “Upscaling”

-Table 1. As authors include the units of conversion (%) and retained activity (%) in the row heading they do not repeat in the next rows.

-KP buffer??-----please include IUPAC NOMENCLATURE

-In the chapter “2.4. Reusability of immobilized hdCK”,  the authors should change conversion percentages by retained activity percentages. Figure 3 should be replaced by supplementary Figure 3.

- Authors said "Scale-up", but they performed the reactions at 12 mM. It can not be reffered as scale up 

MATHERIALS AND METHODS

-Authors needs to include the extinction coeeficient at 280 nm

-E1% value is not the proper term to define the extinction coeeficient (use the greek letter)

DISCUSSION

As human deoxycytidine kinase has been extensively described and characterized, authors can discuss the effect of the immobilization process on enzyme (using PDB structures).

Authors only said that “explanation for reduced activity might be mass-transfer limitations of substrate to contact the enzyme (Sitanggang et al., 2015)”. It is a very poor and non rigurous discussion. Moreover the discussion about the activtity loss after the second reuse is also very limited.

Author Response

Reviewer’s comment: Despite authors have answered several suggestions, the manuscript have some grammar mistakes. In addition, the biocatalyst is previously described (it is not a new enzyme), the immobilization process is not optimized, the discussion is poor, the reusability of the biocatalysts is limited to 1 cycle, and the "scale up" is performed at 12 mM substrate concentration in a final volume of 0.1 ml. Because of this, the impact of this research is far from its potential. 

Answer: Thank you for your comments to our manuscript. It´s correct that the enzyme applied in our study is well described in literature. The broad literature knowledge on the substrate spectrum of human dCK was the reason why we used the enzyme for this study. Enzymes with a broad substrate spectrum are of great importance for synthesis applications. Since a much wider substrate spectrum has been described for human dCK than for other dNKs (e.g. DmdNK), it is a very interesting biocatalyst. To our knowledge human dCK was not used before for the synthesis of high-added value compounds such as what was studied in the manuscript (monophosphate of cladribine, clofarabine and fludarabine).

We see potential to further improve our manuscript based on the reviewers comments, however, within the short time-frame given it will not be possible to gain further experimental data.

Some minor comments:

Reviewer’s comment: Authors should send 2 manuscript versions, the first one with corrections and the second one without any marks, to render the text easier to read.

Answer: A version without comments is now also submitted.

ABSTRACT

Reviewer’s comment: Authors should used the term “Ribonucleoside-5’-monophosphates” instead “ribonucleoside monophosphates” in the different sections of the manuscript.

Answer: The changes were added as suggested.

Reviewer’s comment: Authors mentions that “The immobilized enzyme  was successfully re-used”, however after the second reuse the enzyme loss more than 90 % of the activity. In this sense, authors should remove this claim this.

Answer: The claim was removed from the abstract.

Reviewer’s comment: In addition, they also need to explain the reason of this activity loss.

Answer: We apologize that we did not make the explanation clear enough. We assume that the reason for loss of activity is a detachment of protein over time, which is also described in literature while using ionic interaction for immobilization. While the enzyme remained mainly attached in the first reaction, it was detached during the second reaction. Hence, only few enzyme was transferred to the second reaction.

Further text was added to give further explanation.

Reviewer’s comment: Authors must replace hdCK by HsdCK

Answer: The enzyme name was changed.

RESULTS

Reviewer’s comment: Figure 1. “Immobilization of human dCK to Ni-NTA superflow via ionic interaction” must be replaced by ………………affinity interaction”

Answer: The text was changed as suggested.

Reviewer’s comment: Line 408 “Upsclaing” must be replaced by “Upscaling”

Answer: The heading was changed based on a later suggestion.

Reviewer’s comment: Table 1. As authors include the units of conversion (%) and retained activity (%) in the row heading they do not repeat in the next rows.

Answer: The unit was removed from the table and only kept in the heading.

Reviewer’s comment: KP buffer??-----please include IUPAC NOMENCLATURE

Answer: KP was replaced with potassium phosphate within the whole manuscript. The IUPAC name was given in the materials and methods section.

Reviewer’s comment: In the chapter “2.4. Reusability of immobilized hdCK”,  the authors should change conversion percentages by retained activity percentages. Figure 3 should be replaced by supplementary Figure 3.

Answer: The change was performed as suggested.

Reviewer’s comment: Authors said "Scale-up", but they performed the reactions at 12 mM. It cannot be reffered as scale up 

Answer: The text was changed.

MATHERIALS AND METHODS

Reviewer’s comment: Authors needs to include the extinction coeeficient at 280 nm

Answer: The extinction coefficient was added.

Reviewer’s comment: E1% value is not the proper term to define the extinction coeeficient (use the greek letter)

Answer: The text was changed as suggested.

DISCUSSION

Reviewer’s comment: As human deoxycytidine kinase has been extensively described and characterized, authors can discuss the effect of the immobilization process on enzyme (using PDB structures).

Answer: Thank you very much for your suggestion. PDB studies would be interesting to better understand if the active site is available after immobilization on Ni-NTA sepharose. However, in the short time frame given it was not possible to perform meaningful PDB studies.

Reviewer’s comment: Authors only said that “explanation for reduced activity might be mass-transfer limitations of substrate to contact the enzyme (Sitanggang et al., 2015)”. It is a very poor and non-rigorous discussion.

Answer: Further text was added to manuscript to make the impact of mass transfer limitations clearer. Mass-transfer limitations are widely described in literature and two different mechanisms can be responsible: Either the active sites are not accessible anymore due to immobilization or there is diffusional limitation which leads to gradient formation. Hence, lower product concentrations were observed within the matrix.

We would be happy to receive further explanation or input form the reviewer.

Reviewer’s comment: Moreover, the discussion about the activity loss after the second reuse is also very limited.

Answer: We are relating the loses of activity to the detachment of the enzyme from the matrix. This effect is more pronounced for the third reaction because of the respective increased rounds of immobilized enzyme processing. The statement was re-phrased in the text to make our point clearer.

Reviewer 3 Report

The manuscript by Hellendal and colleagues evaluates the potential of human deoxycytidine kinase as a suitable catalyst for the synthesis of ribonucleoside monophosphates.

Unfortunately, this revised version still lacks experimental details and remains highly confusing.

For example, in the sections presented in line 382 (4.2 Expression of hdCK) and line 387 (4.3 Immobilization of hdCK): It is said that enzyme expression is conducted in 50 ml and 2g of cell pellet were obtained. However, it is also said that an immobilization procedure needs 7.5 g of cell pellet. Please explain this apparent discrepancy. I do not understand the experimental procedure. These values regarding the cell pellets are also presented in the novel Figure 1.

The immobilization is now described in details in section 4.3 (line 387). It is said (line 396) that “200 μl of the Ni-NTA slurry (bed volume of 100 μl) was transferred to a 2 ml reaction tube”. But in line 400, it is also said that “(~ 4.5 ml) was loaded to the equilibrated Ni-NTA matrix”. How was conducted the protein loading? Was a 2 ml reaction tube used during the whole protocol?

More generally, how was conducted this whole set of experiments? It is not clear if the authors used a single transforming colony for the whole study, divided in different cell pellets (the so-called experiments) which were further divided in replicates, or different independent cell cultures for each experiment. Please explain. This information is important as it may help in interpreting the different activity results obtained from the various experiments. A scheme detailing the whole procedure would be helpful.

To add more confusion, it is said in the Results section (line 92) that “1.5 g of cell pellet were needed to reach a sufficient matrix load (Supplementary figure 1)”. However, it is reported in the legend of Supplementary Figure 1 that purification of hdCK by Ni-NTA sepharose was conducting using 0.5 mg of pellet for 100 μl of matrix. Please explain.

Lines 129-133 seem to be disconnected from the main text. Please modify.

It is concluded in lines 141-143 that “no impact of reaction durations up to 24h on the activity of the biocatalyst or the stability of the phosphate donor were observed (Supplementary figure 2). However, data in Supplementary figure 2 only indicate that 100% conversion is observed after 24 hours reaction. Only a kinetic analysis would allow such a conclusion. In the experimental conditions used here, a partial enzyme inactivation may also lead to 100% conversion over such a long period.

Line 144: “Using the soluble and immobilized enzyme hdCK, natural deoxynucleosides were efficiently converted to their equivalent 5’-monophosphate”. What exactly do the authors mean? The conversion by the soluble enzyme is 100% but the conversion by its immobilized counterpart is only 20-30%. Is the immobilized enzyme also considered as efficient? Please explain.

Table 1 presents the role of DTT in the activity of hdCK. The authors report “a minor impact” (line 158). Considering the specific activities presented along with the standard deviations, one may wonder if these results are statistically relevant. A statistical analysis may help the authors to conclude regarding the potential effect of DTT.

Authors have introduced a novel column in Table 1 (“Retained activity”). This calculation should be more clearly presented/explained in the text. But more importantly: how do the authors explain the different values of the “retained activity” according to the substrate? How could a particular substrate affect the retained activity?

Lines 288-289: “Enzyme concentrations of 0.03 mg/ml, 0.19 mg /ml and 0.36 mg /ml were applied”. What is the enzyme concentration used for the results presented in Figure 4? Please explain.

The discussion section has been improved. This section could be further improved by shortening the text and being more concise as it reports too many experimental details from the literature.

Author Response

Reviewer´s comment: Unfortunately, this revised version still lacks experimental details and remains highly confusing.

Answer: Further information on the experimental procedure was added and the manuscript was revised to make our points more clear.

Reviewer’s comment: For example, in the sections presented in line 382 (4.2 Expression of hdCK) and line 387 (4.3 Immobilization of hdCK): It is said that enzyme expression is conducted in 50 ml and 2g of cell pellet were obtained. However, it is also said that an immobilization procedure needs 7.5 g of cell pellet. Please explain this apparent discrepancy. I do not understand the experimental procedure. These values regarding the cell pellets are also presented in the novel Figure 1.

Answer: The text was changed and an additional figure was added to supplementary material to make the procedure more clear.

Reviewer’s comment: The immobilization is now described in details in section 4.3 (line 387). It is said (line 396) that “200 μl of the Ni-NTA slurry (bed volume of 100 μl) was transferred to a 2 ml reaction tube”. But in line 400, it is also said that “(~ 4.5 ml) was loaded to the equilibrated Ni-NTA matrix”. How was conducted the protein loading? Was a 2 ml reaction tube used during the whole protocol?

Answer: The text was changed and an additional figure was added to supplementary material to make the procedure more clear.

Reviewer’s comment: More generally, how was conducted this whole set of experiments? It is not clear if the authors used a single transforming colony for the whole study, divided in different cell pellets (the so-called experiments) which were further divided in replicates, or different independent cell cultures for each experiment. Please explain. This information is important as it may help in interpreting the different activity results obtained from the various experiments. A scheme detailing the whole procedure would be helpful.

Answer: For each immobilization experiment a new glycerol stock was used. All three glycerol stocks used in this study were from the same working cell bank which is based on one single colony. The same information was added to the manuscript. Figure 1A now includes all relevant information for one immobilization experiment. The three immobilization experiments were performed in the same manner.

Reviewer’s comment: To add more confusion, it is said in the Results section (line 92) that “1.5 g of cell pellet were needed to reach a sufficient matrix load (Supplementary figure 1)”. However, it is reported in the legend of Supplementary Figure 1 that purification of hdCK by Ni-NTA sepharose was conducting using 0.5 mg of pellet for 100 μl of matrix. Please explain.

Answer: Supplementary figure 2 shows preliminary results, where the amount of cell pellet needed was optimized. In first experiments 0.5 mg of cell pellet was loaded to the matrix, however, only low amounts of protein were loaded to the matrix. Hence, for further experiments 1.5 mg of cell pellet was loaded to the matrix. Furthermore, the volume of elution buffer was reduced. Further text was added to the manuscript to make it clearer.

Reviewer’s comment:  Lines 129-133 seem to be disconnected from the main text. Please modify.

Answer: It was modified.

Reviewer’s comment:  It is concluded in lines 141-143 that “no impact of reaction durations up to 24h on the activity of the biocatalyst or the stability of the phosphate donor were observed (Supplementary figure 2). However, data in Supplementary figure 2 only indicate that 100% conversion is observed after 24 hours reaction. Only a kinetic analysis would allow such a conclusion. In the experimental conditions used here, a partial enzyme inactivation may also lead to 100% conversion over such a long period.

Answer: Thank you for this helpful comment. You are right, only kinetic analyses provide information about inactivation. Unfortunately, kinetic data can no longer be recorded within the time available. However, the text has been modified.

Reviewer’s comment: Line 144: “Using the soluble and immobilized enzyme hdCK, natural deoxynucleosides were efficiently converted to their equivalent 5’-monophosphate”. What exactly do the authors mean? The conversion by the soluble enzyme is 100% but the conversion by its immobilized counterpart is only 20-30%. Is the immobilized enzyme also considered as efficient? Please explain.

Answer: The text was changed to give a more distinguished picture to:  “Using soluble and immobilized HsdCK, natural deoxynucleosides were converted to their equivalent 5’‑monophosphate except for thymidine and deoxyuridine (Figure 2A). These two were not accepted as a substrate by HsdCK. While the percentage of conversion was approximately 100% using soluble HsdCK, it was between 20% to 40% with the immobilized biocatalyst.”.

Reviewer’s comment: Table 1 presents the role of DTT in the activity of hdCK. The authors report “a minor impact” (line 158). Considering the specific activities presented along with the standard deviations, one may wonder if these results are statistically relevant. A statistical analysis may help the authors to conclude regarding the potential effect of DTT.

Answer: Thank you for your comments on the impact of DTT. Results in the presence and absence of DTT were confusing. Therefore, we performed preliminary studies on the impact of DTT on enzyme attachment to Ni-NTA sepharose. We performed re-usability studies and results indicate that in the presence of DTT HsdCK is immediately detached from the matrix as the biocatalyst could not be re-used at all. Therefore, all results connected to DTT were deleted from the recent manuscript.

Reviewer’s comment:  Authors have introduced a novel column in Table 1 (“Retained activity”). This calculation should be more clearly presented/explained in the text. But more importantly: how do the authors explain the different values of the “retained activity” according to the substrate? How could a particular substrate affect the retained activity?

Answer: It´s difficult to find an answer on this question. We assume that due to the different chemical characteristics of the substrate the diffusion of substrate into the matrix is influenced. While clofarabine seems to easily enter the matrix it is much more difficult for fludarabine. However, further studies are necessary to better understand the observations.

Reviewer’s comment: Lines 288-289: “Enzyme concentrations of 0.03 mg/ml, 0.19 mg /ml and 0.36 mg /ml were applied”. What is the enzyme concentration used for the results presented in Figure 4? Please explain.

Answer: These are the enzyme concentrations applied for Figure 4.

Reviewer’s comment: The discussion section has been improved. This section could be further improved by shortening the text and being more concise as it reports too many experimental details from the literature.

Answer: The discussion was shortened and focused to few important points.

Round 3

Reviewer 3 Report

The manuscript should be now accepted for publication.

Some minor comments:

Fig 4 : Please modify the Figure title "Impact of an increasing substrate concentration..." since both substrate and enzyme concentration are modified in this experience.

Fig S1A: In "Expression of biocatalyst in E. coli BL21", it is written "2 g cell pellet" while in "Cell disruption", it is written "7.5 g of cell pellet"Please modify.

Author Response

The manuscript should be now accepted for publication.

Thank you very much for the numerous hints that have significantly improved our manuscript.

Some minor comments:

Reviewer´s comment: Fig 4 : Please modify the Figure title "Impact of an increasing substrate concentration..." since both substrate and enzyme concentration are modified in this experience.

Answer: The text was changed as suggested.

Reviewer´s comment: Fig S1A: In "Expression of biocatalyst in E. coli BL21", it is written "2 g cell pellet" while in "Cell disruption", it is written "7.5 g of cell pellet"Please modify.

Answer: Figure S1A was modified to make the procedure clearer.